# Neuroprotective and Anti-Amyloid β Effect and Main Chemical Profiles of White Tea: Comparison Against Green, Oolong and Black Tea

**DOI:** 10.3390/molecules24101926

**Published:** 2019-05-19

**Authors:** Xinlei Li, Scott D. Smid, Jun Lin, Zhihong Gong, Si Chen, Fangning You, Yan Zhang, Zhilong Hao, Hongzheng Lin, Xiaomin Yu, Xinyi Jin

**Affiliations:** 1College of Horticulture, Fujian Agriculture and Forestry University, Fuzhou 350002, China; lxlfafu@163.com (X.L.); cstc1990@hotmail.com (S.C.); youfangning123@163.com (F.Y.); zy9531615@163.com (Y.Z.); haozhilong@126.com (Z.H.); linhongzheng2010@126.com (H.L.); 2FAFU-UCR Joint Center for Horticultural Biology and Metabolomics, Fujian Provincial Key Laboratory of Haixia Applied Plant Systems Biology, Fujian Agriculture and Forestry University, Fuzhou 350002, China; realnadal@163.com (J.L.); zhihong_gong@sina.com (Z.G.); 3Discipline of Pharmacology, School of Medical Sciences, Faculty of Health Sciences, The University of Adelaide, South Australia 5000, Australia; scott.smid@adelaide.edu.au; 4Fujian Institute of Microbiology, Fuzhou 350007, China

**Keywords:** white tea, PC-12 cell, Alzheimer’s disease, amyloid β, oxidative stress, UPLC-QTOF-MS, UPLC-QqQ-MS, chemical profiles

## Abstract

White tea (WT) is one of six tea types originally derived from Fujian Province, China. White tea is known for its health-promoting properties. However, the neuroprotective and anti-aggregatory properties of WT against the hallmark toxic Alzheimer’s protein, Aβ have not been investigated. In this study, WT, green tea (GT), oolong tea (OT) and black tea (BT) were manufactured using tea leaves from the cultivar *Camellia sinensis* (Jin Guanyin). The protective effects of these tea extracts were then studied under oxidative stress conditions via *t*-bhp and H_2_O_2_ exposure, in addition to Aβ treatment using a PC-12 cell model. Each tea type failed to rescue PC-12 cells from either *t*-bhp or H_2_O_2_-mediated toxicity, however each extract exerted significant protection against Aβ-evoked neurotoxicity. Results of the Thioflavin T Kinetic (ThT) and TEM assay showed that Aβ aggregate formation was inhibited by each tea type. Additionally, TEM also supported the different anti-aggregatory effect of WT by modifying Aβ into an amorphous and punctate aggregate morphology. Higher accumulated precedent or potential neuroprotective compounds in WT, including ECG’’3Me, 8-*C*-ascorbyl-EGCG, GABA and Gln, in addition to flavonol or flavone glycosides detected by using UPLC-QTOF-MS and UPLC-QqQ-MS, may contribute to a favourable anti-aggregative and neuroprotective effect of WT against Aβ.

## 1. Introduction

Alzheimer’s disease (AD) pathology is characterized by Aβ protein aggregation and senile plaque accumulation in the brain [1]. Aβ peptide is neurotoxic and plays a role in neuronal cell death in AD [2]. Tea is one of most popular beverages globally and has been ascribed a number of health benefits [3]. The relationship between exposure to tea extracts and neuronal protective effects has been widely investigated in both in vivo and in vitro models [4,5]. Many compounds in tea have been found to have neuroprotective effects in vitro associated with detoxifying Aβ fibrils and aggregates, such as catechins, flavonol glycosides and gallic acid, amongst others [6,7,8,9]. Clinical studies have also revealed that people who consume tea may have a slower decline in cognitive function [10,11,12].

According to the different manufacturing methods, tea can be divided into six types: white tea (WT), green tea (GT), oolong tea (OT), black tea (BT), yellow tea and dark tea. WT is a type mainly produced in Fujian, China which has the least processing steps comprising only withering and drying. The feature process of WT is prolonged withering, which reduces the moisture content from 75% in the tea leaves to 20% and allows slight fermentation by endogenous enzymes [13,14]. Studies have already uncovered some relatively potent bioactivities of WT, such as antioxidant [15] and anti-inflammatory actions [16]. However, whether white tea can protect neuronal cells from Aβ-evoked toxicity has not been investigated.

In this study, one batch of tea leaves from *Camellia sinensis* (Jin Guanyin) was used to produce GT, OT, BT, and WT, the four main tea types produced in Fujian Province, China. Neuronal cell protective effects of the tea extracts were then investigated under oxidative stress conditions (*t*-bhp and H_2_O_2_) or following neurotoxic Aβ exposure. In addition, the anti-amyloid effect of each tea was compared using the ThT assay of Aβ fibrillization kinetics and TEM was used to investigate the physical structure of Aβ aggregates. Finally, analyses by UPLC-QTOF-MS and UPLC-QqQ-MS, in combined with chemometrics, were used to compare leaf constituents among the four tea types and dried leaves (DL) to provide insight regarding potential neuroprotective compounds.

## 2. Results

### 2.1. WT Significantly Inhibited Aβ-Evoked Neuronal Cell Death but not Oxidative Stress

To compare the neuronal cell protective effects of different tea types, the same batch of fresh tea leaves were processed into GT, OT, BT, WT and DL according to different manufacturing methods (Appendix A). Tea extracts thus produced from each tea type were assayed for the inhibition of Aβ-evoked neuronal cell death and oxidative stress. Each tea extract failed to rescue PC-12 neuronal cells from either *t*-bhp (50–400 µM) or H_2_O_2_ (400 µM) evoked toxicity (Figure 1 and Figure 2). Instead, all tea extracts increased the toxicity in response to these oxidative stressors, especially at higher concentrations. BT showed less neuronal cell toxicity at higher concentrations, since it did not significantly lower cell viability compared with other tea types (Figure 1C and Figure 2C).

When incubated with 0.5 µM Aβ, PC-12 neuronal cell viability was decreased to 80% compared with PBS treated control cells. Pretreatment with all tea extracts significantly inhibited the reduction in cell viability from Aβ exposure (Figure 3).

### 2.2. Effects of WT on Aβ Fibril Kinetics and Aggregate Morphology (ThT and TEM Assays)

The ThT assay was used to reflect the abundance of β-sheets during fibrillization, which is considered one of the key toxic conformations of the Aβ protein. The relative fluorescence of ThT, indicative of active Aβ fibril formation, rapidly increased in the first 5 h, and thereafter progressed incrementally over the 48 h incubation period (Figure 4A). Similar to other tea types, pretreatment with WT resulted in a marked inhibition of Aβ fibrillization over the course of incubation (Figure 4A,B). This was reflected in the area under the curve results, which displayed a greatly reduced overall ThT fluorescence over time in the presence of each of the tea extracts (Figure 4C).

Further direct Aβ fibril and aggregate observation was conducted using TEM. Aβ alone showed a preponderance of fibrils and aggregates formed after 48 h incubation (Figure 5A). Aβ incubated with GT provided very few areas of fibril density (Figure 5B). OT and BT incubated samples also showed some reduction in overall density of aggregate. However, the aggregate morphology was similar to control samples (Figure 5C,D). Interestingly, after WT treatment Aβ was modified into an amorphous and punctate morphology, which indicated a distinct form of inhibition of fibril and aggregate formation not observed with the other tea extracts (Figure 5E).

### 2.3. UPLC-QTOF-MS and UPLC-QqQ-MS Results Showed That Catechin Derivatives, Flavonol or Flavone Glycosides and Amino Acids were Characteristic Compounds of WT

UPLC-QTOF-MS was employed to provide a comprehensive chemical profiling of each tea type. After filtering, a total of 381 single molecular features were included for PCA. There were 41 compounds identified by comparing their accurate masses, MS/MS fragmentation patterns and UV absorbance with standard compounds or reference standards (Table 1). The identified compounds were further divided into catechins, catechin derivatives, proanthocyanidins, flavonol or flavone glycosides, phenolic acids, hydrolysable tannins, alkaloids and amino acids. The PCA score plot (Figure 6A) showed that in the PC1 axis, WT was located between OT and BT, while in the PC2 axis WT was separated with other tea types. Loading plot of PCA (Figure 6B) results indicated that flavonol glycosides and catechin derivatives were the characteristic compounds in WT. UPLC-QqQ-MS was also used to give a more accurate content of important compounds such as catechins, que-3-rut, amino acids, and caffeine. 

Heatmap data (Figure 7) combined with UPLC-QTOF-MS and UPLC-QqQ-MS results demonstrated that catechin derivatives, flavonol or flavone glycosides and amino acid levels in WT were relatively high. Gallated catechins such as EGCG and ECG have been shown to exert marked anti-aggregatory effects against Aβ when compared to other naturally occurring polyphenols [5]. Although EGCG and ECG levels in WT were lower when compared to GT, they were not significantly different when compared with OT (Figure 7 and Appendix A). Two catechin derivatives in WT, ECG3′’Me and 8-*C*-ascorbyl-EGCG, were found in significantly higher levels than in other tea types (Figure 7 and Appendix A). A majority of flavonol or flavone glycosides were also found in higher levels in WT, especially que-3-glu-rut, myr-3′-glu, myr-3-rob or 3-neo and eri 5,3′-di-glu (Figure 7 and Appendix A). Except for theanine and Met, amino acids levels were also highest in WT (Figure 7 and Appendix A). Gallic acid and theaflavins showed similar tendencies, in that in WT they were found in significantly higher levels than in GT, but were lower than in BT (Figure 7 and Appendix A).

## 3. Discussion

WT and other tea extracts failed to rescue PC-12 cells from toxicity mediated by exposure to the oxidative stressors *t*-bhp and H_2_O_2_. These results are consistent with the effects of major constituent neuroprotective bioactive compounds in tea, such as EGCG, as shown from previous studies [4]. Although many studies have demonstrated the antioxidant activity of tea and its catechins *in vitro* [17,18,19], the time exposed to oxidative stress is usually short (a few hours). Longer times of exposure to ROS, as in the current study, may be more effective at mimicking the constant ROS toxicity occurring in AD-affected brain neurons. Furthermore, EGCG has also been reported to induce cell death via oxidative stress through H_2_O_2_-dependent T-type Ca^2+^ channel opening, or by cell autophagy [20,21]. 

However, WT and other tea extracts significantly prevented the loss of cell viability following Aβ treatment. ThT and TEM results also further supported the anti-aggregative ability of each tea type. TEM further supported that the most effective inhibition of the aggregation effect of WT and GT is by modifying Aβ into seldom aggregate morphology. However, only under WT treatment does the Aβ morphology became amorphous and punctate, indicating a different anti-aggregative effect of WT under TEM observation. 

Distinct chemical profiles in tea which have been reported to possess Aβ anti-aggregative or neuroprotective effects mostly comprise catechins, theaflavins, amino acids and flavonol or flavone glycosides [5,6,22,23]. Catechins are representative of the main chemical profiles of tea, which usually account for 30–42% of the dry weight, while EGCG and ECG are the two main polyphenols that account for 50–80% of total tea catechins. Studies have shown that gallated catechins like EGCG and ECG were able to protect neuronal cells by inhibiting Aβ aggregation, but this does not extend to non-gallated catechins like EC or EGC [4]. Further reports demonstrated that EGCG converts toxic Aβ into SDS-stable, off-pathway and non-toxic oligomers by rotating the galloyl moiety in the B-ring and D-ring [24,25,26]. Interestingly, EGCG and ECG levels in WT were not significantly different with OT, but non-gallated catechins were. This is similar to a previous study that found whole WT processing decreased EGCG and ECG levels by 10% when compared to fresh leaves, while EC, EGC, C and GC decreased by 50%, 40%, 60% and 60%, respectively [27]. Although EGCG content in WT was higher reserved than non-gallated catechin, it was still lower than that in GT and OT. Although EGCG and ECG retained in WT may contribute to its anti-aggregative ability, they may not the key compounds in WT that differentiate the neuroprotective activity of WT from other tea types. It should be noted that although BT was comparatively deficient in the gallated catechins notable for their neuroprotective effects and inhibition of amyloid β aggregation, it still afforded the same degree of neuroprotection as other tea types. This may be due to the higher levels of theaflavin gallates and free gallic acid, which also share similar neuroprotective and anti-amyloid properties [5,6,28]. This may serve to highlight the general health benefits of broader tea consumption generally.

EGCG or ECG in tea can also form derivatives or oxides into catechin derivatives or theaflavins [29,30]. ECG‘’3Me and 8-*C*-ascorbyl-EGCG are two catechin derivatives that have similar structures with their catechin precursors, and were shown to have the highest levels in WT (Figure 7 and Appendix A). Catechin derivatives with new structures and activities have been occasionally reported from diverse tea cultivars or tea types [31,32]. Interestingly, in a recent study, two *N*-ethyl-2-pyrrolidinone-substituted catechin derivatives were found in aged white teas for the first time [33]. They possess similar structures with their catechin precursors. Whether they are also relevant to inhibiting Aβ aggregation by WT requires further experimentation. We speculate that the high content of catechin derivatives in WT may play a key role in its special anti-aggregate ability formation. 

Twelve flavonol or flavone glycosides have been identified in different tea types. Reports showed that many flavonol or flavone glycosides possess anti-aggregative and neuroprotective effects similar to their aglycones [34,35]. A majority of flavonol or flavone glycosides in this study were found at their highest levels in WT. Studies reported that fixed, rotated, fermented and dried steps during tea processing can cause the loss of flavonol or flavone glycosides [36,37,38]. Thus, the accumulation of flavonol or flavone glycosides in WT may also render this tea type with a favorable flavonoid composition conferring anti-amyloid and neuroprotective effects.

Many amines and amino acids were found at their highest levels in WT, including GABA and Gln. GABA is a major inhibitory neurotransmitter in the central nervous system, while GABA derived from natural products has neuroprotective actions [23]. GABA normally occurs in plants at low levels, but increases following exposure to a range of stressors [39,40]. GABA has been shown to protect PC-12 cells from kainic acid excitotoxicity by depressing caspase-3 expression [41]. Additionally, Aβ has been found to elicit GABA-A receptor endocytosis [42] and GABA-A receptor subunit loss is found in AD-affected brains [43]. GABA can also downregulate Aβ-evoked endocytosis to protect neuronal cells [44]. Gln has been reported to activate expression of Hsp70 to protect PC-12 cells from α-synuclein toxicity [45]. Hsp70 activation can also partly mitigate Aβ-evoked cell toxicity [46], while Neuro 2A cells low in Gln are more sensitive to the neurotoxic effects of Aβ [47] and Gln conjugated nanoparticles inhibit amyloidogenesis [48]. Therefore, accumulation of amine-containing compounds like GABA and glutamine in WT may provide a basis for the varying anti-aggregative effect not seen in other tea types.

## 4. Materials and Methods

### 4.1. Chemicals

Human Aβ_1–42_ protein was obtained from rPeptide (Bogart, GA, USA) and Merck Millipore (Bayswater, VIC, Australia). ThT, MTT, Trypan Blue, DMSO, RPMI-1640 medium and FCS were obtained from Sigma-Aldrich (St. Louis, MO, USA). NEAA, penicillin/streptomycin, 10 trypsin EDTA and PBS at pH 7.4 were obtained from Thermo Fisher Scientific (Scoresby, VIC, Australia).

EGCG, EGC, C, ECG, EC, GC, GCG, que-3-glu, gallic acid and amino acids (all with purity ≥ 95%) were obtained from Sigma-Aldrich (St. Louis, MO, USA). EGCG3’’Me (≥95%) and kae-3-rut (≥98%) were purchased from ChemFaces (Wuhan, China). Acetonitrile (MS grade) and methanol (HPLC grade) were obtained from Sigma-Aldrich. Deionized water was produced by a Milli-Q water purification system (Millipore, Billerica, MA, USA).

### 4.2. Tea Samples Preparation for Cell Culture and Chemical Analysis

Twenty-five kilograms of fresh tea leaves of the *Camellia sinensis* (Jin Guanyin) variety (one bud with two or three leaves) were collected from the tea garden at Qilin Mountain Tea Factory, Fujian, China. These tea leaves were then divided into five portions for the manufacturing of DL, GT, OT, BT and WT at the tea factory of Fujian Agriculture and Forestry University by the manufacturing procedures illustrated in Appendix A.

For cell culture experimentation, 25 g of dried tea powder were extracted with water for 30 min at 80 °C and then freeze-dried into powder after filtering. The tea extracts were stored at −80 °C and diluted by PBS until use.

Chemical profiling analysis was performed according to previous methods [49,50]. Briefly, freeze-dried tea leaves were individually ground to fine powders using precooled mortar and pestle. Following lyophilization, 30 mg (±0.5 mg) of ground samples were weighed and 1.2 mL of 70% (*v*/*v*) methanol was added for metabolite extraction. Samples were vortexed, sonicated at 25 °C for 20 min and centrifuged (10 min, 12,000 g). Supernatants were diluted 50-fold with 70% (*v*/*v*) methanol, filtered through a 0.22 µm PVDF filter (Millipore) and stored at −20 °C until analyzed. Three biological sample replicates were prepared for each tea types.

### 4.3. Cell Culture

PC-12 cells (Ordway) displaying a semi-differentiated phenotype with neuronal projections were kindly donated by Professor Jacqueline Phillips (Macquarie University, NSW, Australia) [51]. Cells were maintained in RPMI-1640 media with 10% FCS, 1% Gln, 1% NEAA and 1% penicillin/streptomycin. Cells were seeded at 2 × 10^4^ cells per well in RPMI-1640 with 10% FCS. PC-12 cells were equilibrated for 24 h before treatment with test compounds and/or Aβ.

### 4.4. Cell Treatment and Cell Viability Measurements

Ninety-six well plates containing 2 × 10^4^ PC-12 cells per well were treated with each of the tea extracts (50, 100 or 150 µg/mL) or vehicle (PBS) in appropriate wells. The plates were then incubated for 15 min at 37 °C with 5% CO_2_, with or without tea extracts. Plates were then incubated with either *t*-bhp (50-400 μM) or H_2_O_2_ (400 µM) for 24 h respectively (*n* = 4 each), or with Aβ (0.5 µM) for 48 h (*n* = 6). The concentration setting of *t*-bhp, H_2_O_2_ and Aβ were according to previous studies used in PC-12 cells model [4,52,53].

PC-12 cell viability was determined using the MTT assay. After incubation, removed culture media was replaced with serum-free media containing 0.25 mg/mL of MTT. The plate was further incubated for 2 h, MTT solution removed and cells lysed with DMSO. Absorbance was measured at 570 nm using a Synergy MX microplate reader (Bio-Tek, Bedfordshire, UK).

### 4.5. Aβ Preparation

Native, monomeric pre-fibrillar Aβ was prepared by dissolving in 1% DMSO to yield a protein concentration of 3.8 mM. Sterile PBS was added to prepare a final concentration of 100 μM. Aβ protein was then dispensed into aliquots and immediately frozen at −80 °C until required. 

### 4.6. ThT Assay and TEM of Aβ Fibril and Aggregate Formation

ThT (10 µM in PBS) was added to wells on a black microplate with or without Aβ and tea extracts (100 μg/mL, *n* = 4). Fluorescence was then measured at 37 °C every 10 min for 48 h using a microplate reader (Bio-Tek, Bedfordshire, UK) with excitation and emission wavelengths at 446 nm and 490 nm respectively. ThT output from all treatment groups was normalised to appropriate blanks (ThT with or without tea extracts). 

TEM samples were prepared by incubating native Aβ with or without tea extracts (100 μg/mL) for 48 h at 37 °C. A 400 mesh formvar carbon-coated nickel electron microscopy grid (Proscitech, Kirwan, QLD, Australia) was used. A 5 µL sample was placed onto this grid and after 1 min this sample was blotted off using filter paper. Ten microliters of contrast dye containing 2% uranyl acetate was then placed onto the grid, left for one minute and blotted off with filter paper. Grids were then loaded onto a specimen holder and then into a FEI Tecnai G2 Spirit Transmission Electron Microscope (FEI, Milton, QLD, Australia). Sample grids were then viewed using a magnification of 34,000–92,000×. Grids were extensively scanned manually in search of fibrils and representative images were taken.

### 4.7. UPLC-QqQ-MS Based Targeted Quantification of Catechins, Que-3-Rut, Caffeine and Amino Acids in Different Tea Types

Two microliters of tea extracts or range of the calibration curve were injected on a Waters Acquity UPLC system with Waters photodiode array detector and a XEVO TQ-S MS triple quadrupole mass spectrometer (Waters, Milford, MA, USA).

To detect catechins, que-3-rut and caffeine, chromatographic separation was achieved on a Waters Acquity UPLC BEH C18 column (2.1 × 100 mm, 1.7 µm) at 40 °C with water containing 0.1% formic acid (phase A) and acetonitrile containing 0.1% formic acid (phase B) for chromatographic elution: as previously described [54]. The flow rate was set at 0.3 mL/min. Mass spectrometry was performed in the ESI^-^ for catechins and que-3-rut, ESI^+^ for caffeine under the same settings as previously described [54]. Collision energy and cone voltage were optimized for above compounds with multiple reaction monitoring for quantification. Calibration curves generated by injecting increasing concentrations of chemical standards were used to determine the absolute concentrations of catechins, que-3-rut and caffeine. 

Amino acids were detected in the same manner except that the chromatographic separation was achieved on a Merck SeQuant ZIC-HILIC column (2.1 × 100 mm, 5 µm) at 40 °C with water containing 5 mM ammonium acetate (phase A) and acetonitrile containing 0.1% formic acid (phase B) following our previously published protocol [54]. The flow rate was set at 0.4 mL/min. Mass spectrometry was performed in the ESI^+^ mode using the same setting for caffeine. The MassLynx software (version 4.1, Waters, Milford, MA, USA) was used for instrument control and data acquisition.

### 4.8. UPLC-QTOF MS-Based Non-Targeted Metabolite Analysis of Different Tea Types

The metabolomics measurements and analysis were carried out according to our previous method [54]. Briefly, one microliter of the metabolite extract was injected into an Acquity UPLC system coupled in tandem to a photodiode array detector and a SYNAPT G2-Si HDMS QTOF mass spectrometer (Waters, Milford, MA, USA). Separation was achieved on a Waters Acquity UPLC HSS T3 column (2.1 × 100 mm, 1.8 µm) thermostat controlled at 40 °C using a gradient from solvent A (water with 0.1% formic acid) to solvent B (acetonitrile with 0.1% formic acid). The flow rate was set at 0.3 mL/min. Data was collected in the ESI mode, scanning from 50–1200 *m*/*z*. QC samples were prepared by mixing an equal amount of each sample to become a combined sample, and were injected every five samples throughout the runs to monitor the instrument performance. The MassLynx software (version 4.1, Waters, Milford, MA, USA) was used to control of the instruments. Each tea sample was analyzed in triplicates.

### 4.9. Data Processing, Metabolite Identification, and Statistical Analysis

Data obtained from the MTT assay was analyzed via a two-way analysis of variance (ANOVA) to assess neuronal cell viability arising from incubation in H_2_O_2_, *t*-bhp or Aβ, alone or in the presence of tea extracts, with a Bonferroni’s post hoc test used to determine the significance level for each tea extract treatment. Area under the curve analysis for ThT fluorescence data was analyzed using one-way ANOVA with a Dunnett’s multiple comparisons test used for determining the significance of each extract vs. Aβ. Data of chemical profiles of tea was analyzed via Tukey’s HSD test. Resulting chromatograms from UPLC-QTOF MS were processed in Progenesis QI. After filtering, a total of 381 single molecular features were used as inputs for PCA and Loading Plot to observe intrinsic metabolite variances between tea types using Progenesis QI extension EZinfo after Pareto scaling. Differential metabolites among tea samples with VIP >1 and *p* value < 0.05 were identified. Data analysis and production of graphs was performed in GraphPad Prism 6 for Windows (GraphPad Software, San Diego, USA) or SPSS (version 19.0, Chicago, IL, USA). 

## 5. Conclusions

In summary, WT and other tea types failed to protect neuronal cells from oxidative stress. However, each tea type significantly increased neuronal cell viability following Aβ exposure. ThT fluorescence kinetics and TEM observation showed that Aβ aggregate formation was dramatically inhibited by each tea type. TEM also supported that the most effective inhibition of the aggregation effect of WT and GT was by modifying Aβ into seldom aggregate morphology. However, amorphous and punctate Aβ morphology was observed only following WT treatment, indicating a different anti-aggregative effect of WT under TEM observation. Chemical analysis revealed that EGCG and ECG in WT were significantly lower than in GT, but accumulation of other bioactive compounds such as ECG3′’Me, 8-*C*-ascorbyl-EGCG, que-3-glu-rut, myr-3′-glu, myr-3-rob or 3-neo, eri 5,3′-di-glu and other flavonol or flavone glycosides may underlie the anti-aggregative and neuroprotective effect of WT. Particularly, GABA and glutamine levels were much higher in WT than other tea types, and this may also facilitate the neuroprotective effect of this tea type. Further studies are needed to characterize any discrete neuroprotective effects of some of these compounds found in WT and in vitro findings need to be substantiated via appropriate in vivo paradigms.

## Figures and Tables

**Figure 1 molecules-24-01926-f001:**
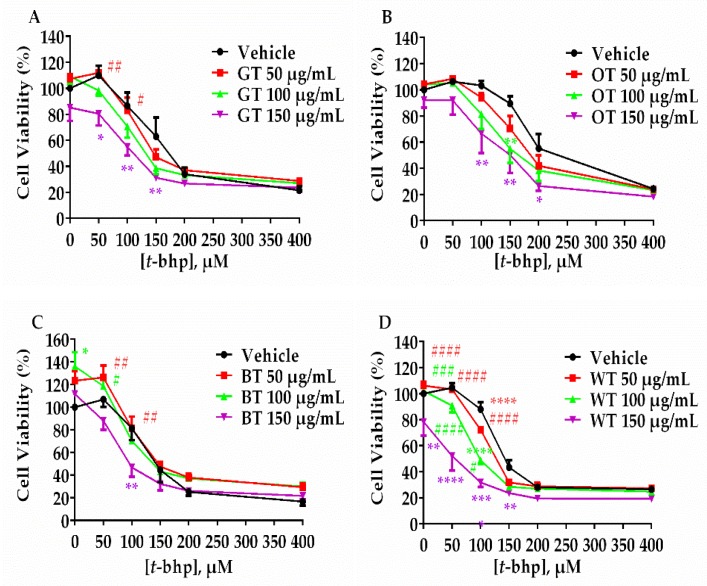
MTT assay of cell viability in response to 24 h incubation with *t*-bhp (0–400 µM), alone or in the presence of (**A**) GT, (**B**) OT, (**C**) BT and (**D**) WT extracts (50, 100 or 150 μg/mL). * *p* < 0.05, ** *p* < 0.01, *** *p* < 0.001, **** *p* < 0.0001 vs. vehicle. ^#^
*p* < 0.05, ^##^
*p* < 0.01, ^###^
*p* < 0.001, ^####^
*p* < 0.0001 vs. 150 μg/mL. Results were expressed as mean ± standard deviation (*n* = 4).

**Figure 2 molecules-24-01926-f002:**
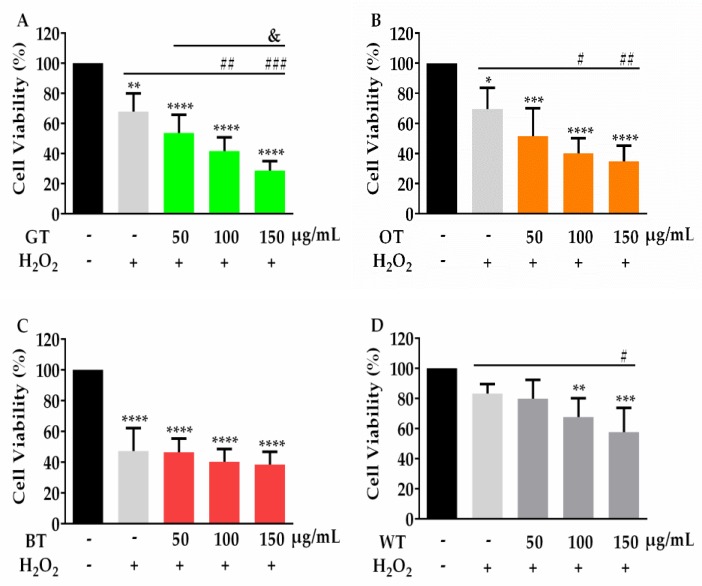
MTT assay of neuronal PC12 cell viability in response to 24 h incubation with H_2_O_2_ (400 µM), alone or in the presence of (**A**) GT, (**B**) OT, (**C**) BT and (**D**) WT extracts (50, 100 or 150 μg/mL). * *p* < 0.05, ** *p* < 0.01, *** *p* < 0.01, **** *p* < 0.0001 vs. vehicle. ^#^
*p* < 0.05, ^##^
*p* < 0.01, ^###^
*p* < 0.001 vs. H_2_O_2_ alone. ^&^
*p* < 0.05 vs. H_2_O_2_ + 50 μg/mL tea extracts. Results were expressed as mean ± standard deviation (*n* = 4).

**Figure 3 molecules-24-01926-f003:**
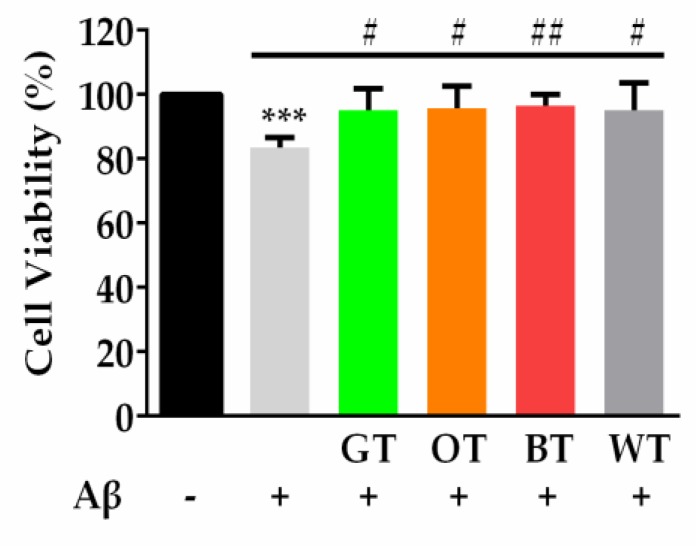
MTT assay of cell viability in response to 48 h incubation with Aβ (0.5 µM), alone or in the presence of each tea extract (100 μg/mL). *** *p* < 0.001, vs. Control. ^#^
*p* <0.05, ^#^^#^
*p* < 0.01 vs. Aβ. Results were expressed as mean ± standard deviation (*n* = 6).

**Figure 4 molecules-24-01926-f004:**
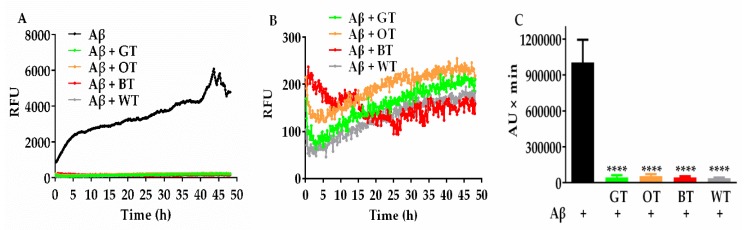
(**A**) Thioflavin T Kinetic (ThT) fluorescence representing kinetics of Aβ (10 µM) fibrillization over 48 h, alone or in the presence of each tea extract (100 μg/mL). (**B**) ThT fluorescence representing kinetics of Aβ (10 µM) fibrillization over 48 h, in the presence of each tea extract (100 μg/mL). (**C**) Respective area under (AU) the curve measurements per minute demonstrating significant reductions in ThT fluorescence output for WT, GT, OT and BT. **** *p* < 0.0001. Results were expressed as mean ± standard deviation (*n* = 4).

**Figure 5 molecules-24-01926-f005:**
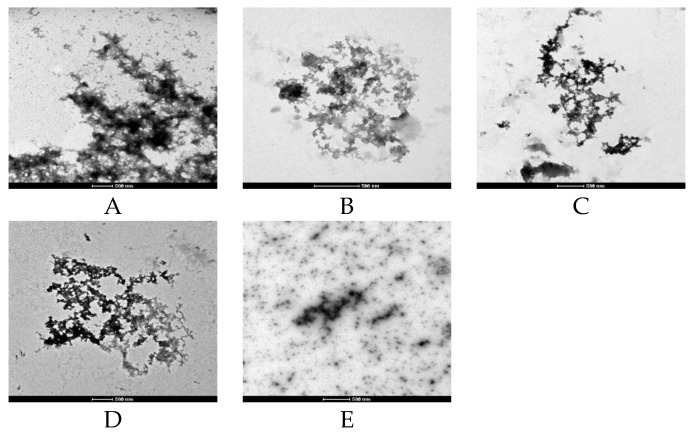
Representative transmission electron micrographs demonstrating effects of each tea extract (100 μg/mL) on Aβ (10 µM) fibril and aggregate formation after 48 h incubation. (**A**) Aβ, (**B**) GT + Aβ, (**C**) OT + Aβ, (**D**) BT + Aβ and (**E**) WT + Aβ. Scale bar: 500 nm (**A**–**E**).

**Figure 6 molecules-24-01926-f006:**
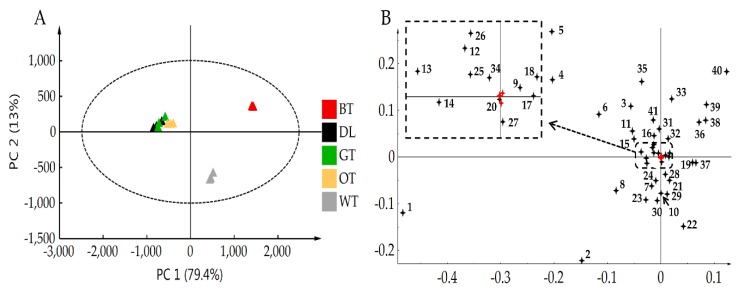
PCA of different tea types. (**A**) Score plot of PCA demonstrating differences in metabolite profiles between leaf samples based on 381 filtered single molecular features detected by UPLC-QTOF-MS in ESI^−^. (**B**) Loading plot of PCA indicating primary differential metabolites. Experiments were performed in triplicates.

**Figure 7 molecules-24-01926-f007:**
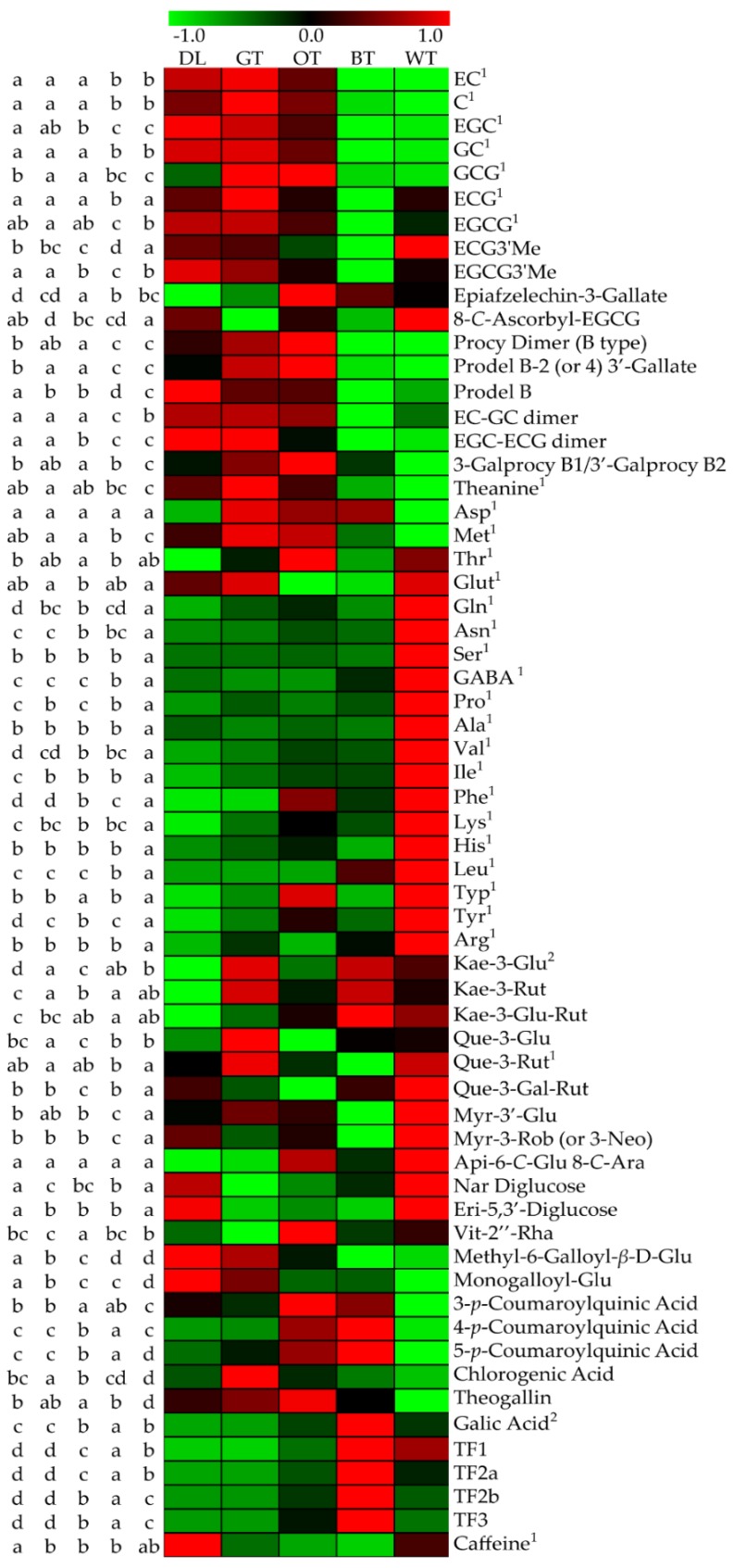
Comparisons of metabolite intensity in different tea types. Normalized values from three biological replicates were shown on a color scale proportional to the content of each metabolite and are expressed as log2. Lowercase letters on the left indicate significant differences (*p* < 0.05) among different samples. Compound name on the right with superscript 1 means they were detected by UPLC-QqQ-MS and superscript 2 means they were confirmed by authentic standards detected by UPLC-QTOF-MS. Others were identified through MS/MS information from UPLC-QTOF-MS.

**Table 1 molecules-24-01926-t001:** Metabolites putatively identified in four tea types by UPLC-QTOF-MS.

Compounds	Tentative Assignment	RT (min)	Detected [M − H]^−^ (*m*/*z*)	Theoretical [M − H]^−^ (*m*/*z*)	Mass Error (ppm)	Formula	MS/MS Fragments
1	EGCG ^a^	6.34	457.0816	457.0771	−3.28	C_22_H_18_O_11_	457.0781, 305.0669, 169.0147
2	ECG ^a^	7.83	441.0832	441.0822	−2.27	C_22_H_18_O_10_	289.0720, 169.0144
3	C ^a^	5.36	289.072	289.0712	−2.77	C_15_H_14_O_6_	245.0816, 203.0709
4	EGC ^a^	4.92	305.0704	305.0661	−3.93	C_15_H_14_O_7_	305.0672, 219.0661
5	EC ^a^	6.25	289.0723	289.0712	−3.81	C_15_H_14_O_6_	245.082, 203.0711
6	GC ^a^	3.85	305.0667	305.0661	−1.97	C_15_H_14_O_7_	219.0659, 125.0242
7	ECG3’’Me	8.88	455.0983	455.0978	−1.10	C_23_H_20_O_10_	290.0750, 183.0298
8	EGCG3’’Me ^a^	7.41	471.0933	471.0927	−1.27	C_23_H_20_O_11_	287.0534, 183.0298
9	Epiafzelechin-3-Gallate	8.93	425.0878	425.0873	−1.25	C_22_H_18_O_9_	125.0243, 137.0245, 273.0761, 169.0140, 151.0029,
10	8-*C*-Ascorbyl-EGCG	6.66	631.0945	631.0935	−1.58	C_28_H_24_O_17_	479.0816, 423.0706, 316.0225
11	Procy Dimer (B type)	5.68	577.1353	577.1346	−1.21	C_30_H_26_O_12_	425.0872, 407.0768, 289.0716
12	Prodel B-2 (or 4) 3’-*O*-Gallate	5.13	761.1356	761.1354	−0.26	C_37_H_30_O_18_	423.0718, 609.1236, 591.1135, 577.1348,
13	Prodel B Isomer 2	4.13	609.1249	609.1244	−0.82	C_30_H_26_O_14_	441.0827, 423.0717, 305.0665
14	EC-GC Dimer	4.81	593.1300	593.1295	−0.84	C_30_H_26_O_13_	423.0712
15	EGC-ECG Dimer	6.03	745.1407	745.1405	−0.27	C_37_H_30_O_17_	593.1300, 407.0767
16	3-Galloyl-Procy B1/3’-Galloyl-Procy B2	6.78	729.1458	729.1456	−0.27	C_37_H_30_O_16_	577.1228, 407.0768, 289.0715
17	Kae-3-Glu ^a^	8.76	447.0925	447.0927	0.45	C_21_H_20_O_11_	255.0298, 227.0348
18	Kae-3-Rut	8.42	593.1509	593.1506	−0.51	C_27_H_30_O_15_	285.0399
19	Kae-3-Glu-Rut	8.00	755.2038	755.2035	−0.40	C_33_H_40_O_20_	285.0404
20	Que-3-Glu	8.00	463.0879	463.0877	−0.43	C_21_H_20_O_12_	300.0275, 271.0252
21	Que-3-Rut ^a^	7.69	609.1445	609.1456	1.81	C_27_H_30_O_16_	271.0248
22	Que-3-Glu-Rut	7.35	771.1988	771.1984	−0.52	C_33_H_40_O_21_	609.1461, 301.0348
23	Myr-3’-Glu	7.11	479.0819	479.0826	1.46	C_21_H_20_O_13_	287.0201, 271.0247
24	Myr-3-Rob (or 3-Neo)	6.93	625.1400	625.1405	0.80	C_27_H_30_O_17_	527.1541, 307.1396
25	Methyl-6-Galloyl-*β*-d-Glu	3.68	345.0818	345.0822	1.16	C_14_H_18_O_10_	285.0611, 225.0401, 183,0296
26	Monogalloyl Glu	2.49	331.0669	331.0665	−1.21	C_13_H_16_O_10_	169.0142
27	Vitexin-2’’-Rha	7.67	577.1551	577.1557	1.04	C_27_H_30_O_14_	323.0557, 282.0533
28	Api-6-*C*-Glu 8-*C*-Ara	6.90	563.1405	563.1401	−0.71	C_26_H_28_O_14_	473.1089, 383.0770, 353.0665
29	Naringenin Diglucose	6.13	595.1660	595.1663	0.50	C_27_H_32_O_15_	475.1245, 433.1347, 313.0925
30	Eri-5,3’-Diglucose	6.05	611.1617	611.1612	−0.82	C_27_H_32_O_16_	491.1194, 449.1288, 329.0868
31	3-*p*-Coumaroylquinic Acid	5.16	337.0930	337.0923	−2.08	C_16_H_18_O_8_	191.0559, 163.0399, 119.0500
32	4-*p*-Coumaroylquinic Acid	6.12	337.0930	337.0923	−2.08	C_16_H_18_O_8_	173.0455, 163.0397, 1193.0502
33	5-*p*-Coumaroylquinic Acid	6.40	337.0931	337.0923	−2.37	C_16_H_18_O_8_	173.0460, 163.0400, 119.0502
34	Chlorogenic Acid	5.51	353.0877	353.0873	−1.13	C_16_H_18_O_9_	191.0558, 179.0349, 173.0455, 135.0449
35	Theogallin	2.94	343.0666	343.0665	−0.29	C_14_H_16_O_10_	343.0670, 191.0562
36	Galic Acid ^a^	2.78	169.0142	169.0137	−2.96	C_7_H_6_O_5_	125.0242, 95.0138
37	TF1	10.57	563.1196	563.1190	−1.07	C_29_H_24_O_12_	425.092
38	TF2a	10.94	715.1303	715.1299	−0.56	C_36_H_28_O_16_	563.119
39	TF2b	11.18	715.1301	715.1299	−0.28	C_36_H_28_O_16_	289.0721, 245.0820
40	TF3	11.12	867.1416	867.1409	−0.81	C_43_H_32_O_20_	867.1414
41	l-Theanine	1.36	173.0932	173.0926	−3.58	C_7_H_14_N_2_O_3_	155.0830, 128.0354

Note: ^a^ This letter indicates that identification of the compound was confirmed by the authentic standard.

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
