# Peer review of "Neuroprotective and Anti-Amyloid β Effect and Main Chemical Profiles of White Tea: Comparison Against Green, Oolong and Black Tea"

_molecules, 2019, doi:10.3390/molecules24101926_

Round 1
Reviewer 1 Report
Expand the introduction, on the white tea.
In discussion, compare the results, with a possible activation of Nrf-2 by the white tea.
What will be the expectations of the job, in the usefulness of white tea.
Author Response
Dear reviewer,
Thank you for your comments on our manuscript. According to your comments, we have revised our manuscript very carefully. All the revised portions have been marked using red text. Here are your comments and our responses:
Point 1: Expand the introduction, on the white tea
Response 1: We have expanded the Introduction, especially in relation to white tea. The increased places were in page 2 line 51-54 and highlighted by red text in new version of manuscript.
Point 2: In discussion, compare the results, with a possible activation of Nrf-2 by the white tea.
Response 2: In our study, we studied the antioxidative effect of tea extract. However, each tea extract failed to rescue PC-12 cells from oxidative stressors. So, we think the pathways involved in the cell oxidative stress including Nrf-2 may not activated in our study, or insufficient to counter t-bhp and H2O2 toxicity. Amyloid beta can generate ROS, so this may occur but be more subtle and be able to be mitigated by Nrf activation in PC-12 cells. Many substances in tea extracts like catechins, theaflavin, flavonol glycosides and gallic acid, amongst others have been reported to interact with amyloid directly to decrease toxicity. Thus, in this study, we emphasised the anti-aggregative effect of tea extracts as a likely mechanism of action
Point 3: What will be the expectations of the job, in the usefulness of white tea.
Response 3: Important aspects of our study have been covered in the current manuscript as follows:
White tea is a type which has the least processing steps to production, that being only withering and drying. White tea nowadays has become increasingly popular in China and around the world. The first expectation of our study is to determine whether white tea has neuroprotective and/or anti-aggregate effects. The second expectation is whether white tea has a greater neuroprotective or anti-aggregative effect compared with green tea, oolong tea or black tea, which are the four typical tea types from Fujian, China. The third expectation is to pinpoint distinct constituents in white tea that are potentially involved in its neuroprotective activities. According to our results, white tea significantly increased neuronal cell viability following Aβ exposure associated with alteration of amyloid into an amorphous and punctate morphology. Chemical analysis also revealed potential neuroprotective compounds like ECG’’3Me, 8-C-ascorbyl-epigallocatechin-3-gallate, Quercetin-3-O-Gulcosyl-Rutinoside, Myricetin-3'- Glucoside, Myricetin-3-Robinobioside (or 3-Neohesperidoside) and Eriodictyol 5,3'-Di-O-Glucoside were abundant in white tea. This study provides insight into white tea and any signature bioactive compounds that may be associated with neuroprotection in types of dementia such as Alzheimer’s Disease. These metabolites will be the focus of our future studies to isolate and test the activities in vitro.

Reviewer 2 Report
The work presented by Xinlei Li et al describes the protective effects of tea extracts under oxidative stress conditions promoted by tert-butyl hydroperoxide (t-bhp) and H2O2exposure, in addition to Aβtreatment using a PC-12 neuronal cell toxicity model.
From these results the authors identified that each tea type failed to rescue PC-12 cells from either t-bhp or H2O2-mediated toxicity. However, each extract exerted significant protection against Aβ-evoked neurotoxicity. Furthermore, transmission electron microscopy supported the different anti-aggregatory effect of 30 WT by modifying Aβ into an amorphous and punctate aggregate morphology.
To sum up, I think the work is novel, but not sound, and the data must be better discussed. Consequently, I suggest to accept it for publication, after minor revision.
Author Response
Dear reviewer,
Thank you for your comments on our manuscript. According to your comments, we have revised our manuscript very carefully. All the revised portions have been marked using red text. Here are your comments and our responses:
Point 1: Data must be better discussed.
Response 1: We have further discussed on our Results in the Discussion, such as comparative component neuroprotective compounds in the varying tea types and precedent studies discussing individual bioactive protective mechanisms. This is included on page 10, lines 179-180; page 11 lines 197-201 & 207-215. All changes in the revised manuscript are highlighted in red text.

Reviewer 3 Report
In this paper, Xinlei and colleagues described the chemical profiles of white, green, oolong and black teas and their neuroprotective and anti-amyloid β effects. Moreover, the Authors showed the protective activity of these four tea extracts in the PC-12 cells under oxidative stress (t-BHP or H2O2) conditions.
The paper fits with the scope of “Molecules” and could be interesting, yet this study does not reach a sufficient level for publication in this journal. This because the biological part is poor, has critical methodological issues and its results are overestimated.
Following, I report some advices to improve the work:
- Authors employed a very high concentration of H2O2 (400 µM), despite its physiological cellular content is in the nanomolar range, and just few µM are toxic. Therefore, Authors should test much lower concentrations of stressors;
- the conclusions about the potential neuroprotective activity of the extracts are overestimated. Indeed, the MTT assay employed to this aim is not enough to demonstrate it. Authors should use more than one test to prove neuroprotection. In other words, the MTT results could be confirmed by another non-colorimetric assay aimed to evaluate the cell viability.
- Authors should search for a possible mechanism of action of the tea extracts.
- Authors should revise the English language.
- in the paragraph 5.4., Authors should indicate the employed concentration of tea extracts and provide more details, such as the number of replicates of each experiment and the number of performed experiments.
- in the captions, it should be specified if the histograms report DS or SEM.
Author Response
Dear reviewer,
Thank you for your comments on our manuscript. According to your comments, we have revised our manuscript very carefully. All the revised portions have been marked using red text. Here are your comments and our responses:
Point 1: Authors employed a very high concentration of H2O2 (400 µM), despite its physiological cellular content is in the nanomolar range, and just few µM are toxic. Therefore, Authors should test much lower concentrations of stressors.
Response 1: We added t-bhp and H2O2 to the extracellular environment, thereby require higher concentrations than what would be required intracellularly. However, these are well established and validated from prior use where we see incremental and concentration-dependent toxicity to a degree required to see any neuroprotection from bioactives. Lower concentrations of H2O2 (<300 µM) we have found are insufficient in their toxicity to observe neuroprotective effects of various bioactives.
Point 2: the conclusions about the potential neuroprotective activity of the extracts are overestimated. Indeed, the MTT assay employed to this aim is not enough to demonstrate it. Authors should use more than one test to prove neuroprotection. In other words, the MTT results could be confirmed by another non-colorimetric assay aimed to evaluate the cell viability.
Response 2: MTT assay is a robust validated and recognised measure of neuronal cell viability via mitochondrial activity. While other assays are available and would be complimentary to the MTT assay, we feel that is sufficient standalone bioassay to measure our desired experimental outcomes, in which we saw statistically significant protective effects of tea extracts on Abeta-mediated cell toxicity.
Point 3: Authors should search for a possible mechanism of action of the tea extracts.
Response 3: Studies have been proved that amyloid can cause neuronal cell death by generating ROS. Thus, in our study, we studied the antioxidative effect of tea extract. However, each tea extract failed to rescue PC-12 cells from oxidative stressors. Since many metabolites in tea extracts like catechins, theaflavin, flavonol glycosides and gallic acid, amongst others have been reported to interact with amyloid directly to decrease its toxicity. We opted to focus on the likelihood that these constituent compounds mediated protective effects through this pathway in our Discussion, but also described more generalised cellular protective effects recruited through other component bioactives (e.g. GABA). These aspects have been enhanced in the revised Discussion.
Point 4: Authors should revise the English language.
Response 4: The English language of this manuscript has been revised where appropriate and highlighted in red text in the revised version of manuscript.
Point 5: in the paragraph 5.4., Authors should indicate the employed concentration of tea extracts and provide more details, such as the number of replicates of each experiment and the number of performed experiments.
Response 5: The concentration of tea extracts and other details have been added in Paragraph 5.4 and in all Figure captions, and highlighted in red text in the revised version of manuscript.
Point 6: in the captions, it should be specified if the histograms report SD or SEM.
Response 6: All error bars in histograms are now referred to in standard deviation (SD) units and this information was added in Fig.2, Fig.3, Fig.4, Fig.5 and Fig.9 captions and highlighted in red text in the revised version of this manuscript.

Round 2
Reviewer 3 Report
Reviewer’s Point 1: Authors employed a very high concentration of H2O2 (400 µM), despite its physiological cellular content is in the nanomolar range, and just few µM are toxic. Therefore, Authors should test much lower concentrations of stressors.
Authors’ Response 1: We added t-bhp and H2O2 to the extracellular environment, thereby require higher concentrations than what would be required intracellularly. However, these are well established and validated from prior use where we see incremental and concentration-dependent toxicity to a degree required to see any neuroprotection from bioactives. Lower concentrations of H2O2 (<300 µM) we have found are insufficient in their toxicity to observe neuroprotective effects of various bioactives.
Reviewer’s Response 1: The Authors’ answer is absolutely lacking, because it is conceptually wrong. The authors do not seem to understand that to recover from 400 µM concentration of H2O2, a miracle would be needed. Very numerous reports claim a strong neurotoxicity already at concentrations lower than 100 µM for an incubation of 24 hours. Hence, I firmly contest this experimental approach!
Reviewer’s Point 2: the conclusions about the potential neuroprotective activity of the extracts are overestimated. Indeed, the MTT assay employed to this aim is not enough to demonstrate it. Authors should use more than one test to prove neuroprotection. In other words, the MTT results could be confirmed by another non-colorimetric assay aimed to evaluate the cell viability.
Authors’ Response 2: MTT assay is a robust validated and recognised measure of neuronal cell viability via mitochondrial activity. While other assays are available and would be complimentary to the MTT assay, we feel that is sufficient standalone bioassay to measure our desired experimental outcomes, in which we saw statistically significant protective effects of tea extracts on Abeta-mediated cell toxicity.
Reviewer’s Response 2: What Authors stated is wrong and extremely insufficient to justify a methodological mistake acknowledged in the whole scientific community. Indeed, it is well-known that employing only one technique to prove whichever effect would be considered insufficient. Moreover, the MTT assay measures indirectly the cell viability via mitochondrial activity, but some chemicals or phytochemicals may change the activity of succinate dehydrogenase or interact with MTT directly, thus interfering with the analysis. Furthermore, the methods employed for the extraction of the plant materials (i.e. solvents) may cause interference in with the assay, resulting in an unreliable viability. Therefore, I absolutely reiterate the need of using more than one test to evaluate cell viability.
Reviewer’s Point 3: Authors should search for a possible mechanism of action of the tea extracts.
Authors’ Response 3: Studies have been proved that amyloid can cause neuronal cell death by generating ROS. Thus, in our study, we studied the antioxidative effect of tea extract. However, each tea extract failed to rescue PC-12 cells from oxidative stressors. Since many metabolites in tea extracts like catechins, theaflavin, flavonol glycosides and gallic acid, amongst others have been reported to interact with amyloid directly to decrease its toxicity. We opted to focus on the likelihood that these constituent compounds mediated protective effects through this pathway in our Discussion, but also described more generalised cellular protective effects recruited through other component bioactives (e.g. GABA). These aspects have been enhanced in the revised Discussion.
Reviewer’s Response 3: I believe that without experimental results explaining any possible mechanism of ation, the paper cannot be accepted in this journal.
Author Response
Dear reviewer,
Thank you for your comments on our manuscript. According to your comments, we have revised our manuscript very carefully. All the revised portions have been marked using red text. Here are your comments and our responses:
Reviewer’s Point 1: The Authors’ answer is absolutely lacking, because it is conceptually wrong. The authors do not seem to understand that to recover from 400 µM concentration of H2O2, a miracle would be needed. Very numerous reports claim a strong neurotoxicity already at concentrations lower than 100 µM for an incubation of 24 hours. Hence, I firmly contest this experimental approach!
Authors’ Response 1:
For H2O2 treatment, we also had tried from lower concentrations (<300 µM) but insufficient in their toxicity to observe neuroprotective effects of various bioactives so we increase the concentration to 400 µM. This concentration setting is similar with previous studies that they used 300-500 µM concentration H2O2 in PC-12 cells model, so we further quoted two references in the revised manuscript in page 12 line 275-276. Thus, we think it might be caused by the different neuronal cell model. In PC-12 cells model, the higher concentration of H2O2 may be needed to cause enough toxicity.
Reviewer’s Point 2: What Authors stated is wrong and extremely insufficient to justify a methodological mistake acknowledged in the whole scientific community. Indeed, it is well-known that employing only one technique to prove whichever effect would be considered insufficient. Moreover, the MTT assay measures indirectly the cell viability via mitochondrial activity, but some chemicals or phytochemicals may change the activity of succinate dehydrogenase or interact with MTT directly, thus interfering with the analysis. Furthermore, the methods employed for the extraction of the plant materials (i.e. solvents) may cause interference in with the assay, resulting in an unreliable viability. Therefore, I absolutely reiterate the need of using more than one test to evaluate cell viability.
Authors’ Response 2:
We understand and agree with that more technique used will increase the reliability and accuracy of the results. However, we think that MTT assay is sufficiency standalone bioassay to measure our desired experimental outcomes. Firstly, MTT assay is a robust validated and recognised measure of neuronal cell viability via mitochondrial activity. Secondly, the H2O2 treatment results was similar with our previous study on EGCG and all kinds of tea extracts failed to protect neuronal cells against oxidative stress in vitro. Finally, t-bhp treatment also proved that the tea extracts cannot prevent the loss of the PC-12 cell viability.
For MTT assay, as demonstrate in 4.4, serum-free media containing 0.25 mg/ml of MTT was added after removing culture media after incubation. Thus, the tea extracts may not able to interact with MTT directly. Furthermore, we observed that all kinds of tea extracts significantly rescue PC-12 cells from Abeta treatment using MTT assay which increased the reliable of MTT assay use in our study.
On the basis of above reasons, we do not used another cell viability assay in this study.
Reviewer’s Point 3: I believe that without experimental results explaining any possible mechanism of action, the paper cannot be accepted in this journal.
Authors’ Response 3:
We firmly agree with that the PC-12 cells will generate ROS and activate some pathways related cell death like caspase under Abeta toxicity. So, in our study, we studied the neuronal cell protective effects of the tea extracts under oxidative stress conditions (t-bhp and H2O2) and Abeta treatment. We found that each tea extract failed to protect neuronal cells from oxidative stress but significantly increased neuronal cell viability following Abeta exposure. Thus, we considered that the key role of tea extract protect PC-12 cells under Abeta exposure in vitro may be the ability to interact with Abeta directly rather than interact with cells. Further ThT and TEM results also supported this point of view. Thus, we think that is not meaningful for us to investigate the protect mechanism of tea extracts related with cells. However, we also agree that the white tea was participated in the cell gene expression regulation to some extent under Abeta toxicity for the high accumulation of GABA or Glutamine. We think further study need to be done to investigate the potential mechanism of action in vitro and in vivo.

Round 3
Reviewer 3 Report
Dear reviewer,
Thank you for your comments on our manuscript. According to your comments, we have revised our manuscript very carefully. All the revised portions have been marked using red text. Here are your comments and our responses:
I round Reviewer’s Point 1: Authors employed a very high concentration of H2O2 (400 µM), despite its physiological cellular content is in the nanomolar range, and just few µM are toxic. Therefore, Authors should test much lower concentrations of stressors.
I round Authors’s Response 1: We added t-bhp and H2O2 to the extracellular environment, thereby require higher concentrations than what would be required intracellularly. However, these are well established and validated from prior use where we see incremental and concentration-dependent toxicity to a degree required to see any neuroprotection from bioactives. Lower concentrations of H2O2 (<300 µM) we have found are insufficient in their toxicity to observe neuroprotective effects of various bioactives.
II round Reviewer’s Response 1: The Authors’ answer is absolutely lacking, because it is conceptually wrong. The authors do not seem to understand that to recover from 400 µM concentration of H2O2, a miracle would be needed. Very numerous reports claim a strong neurotoxicity already at concentrations lower than 100 µM for an incubation of 24 hours. Hence, I firmly contest this experimental approach!
II round Authors’ Response 1: For H2O2 treatment, we also had tried from lower concentrations (<300 μM) but insufficient in their toxicity to observe neuroprotective effects of various bioactives so we increase the concentration to 400 μM. This concentration setting is similar with previous studies that they used 300-500 μM concentration H2O2 in PC-12 cells model, so we further quoted two references in the revised manuscript in page 12 line 275-276. Thus, we think it might be caused by the different neuronal cell model. In PC-12 cells model, the higher concentration of H2O2 may be needed to cause enough toxicity.
III round Reviewer’s Response 1: I reiterate my disappointment for this experimental design! Below are reported some scientific papers showing that the concentrations of H2O2 able to induces toxicity in PC-12 cells are much lower than 300 μM, starting from 25 μM. (e.g. Liu et al., 2019; Niu et al., 2018; Zhang et al., 2018; Luo et al., 2018; Yin et al., 2018). Therefore, conversely to what Authors think, the experiments should be redone with much lower concentrations of stressor.
Liu H, Chen B, Zhu Q. Long non-coding RNA SNHG16 reduces hydrogen peroxide-induced cell injury in PC-12 cells by up-regulating microRNA-423-5p. Artif Cells Nanomed Biotechnol. 2019 Dec;47(1):1444-1451. doi: 10.1080/21691401.2019.1600530.
Niu T, Jin L, Niu S, Gong C, Wang H. Lycium Barbarum Polysaccharides Alleviates Oxidative Damage Induced by H2O2 Through Down-Regulating MicroRNA-194 in PC-12 and SH-SY5Y Cells. Cell Physiol Biochem. 2018;50(2):460-472. doi: 10.1159/000494159.
Zhang X, Li Z, Zhang Q, Chen L, Huang X, Zhang Y, Liu X, Liu W, Li W. Mechanisms Underlying H2O2-Evoked Carbonyl Modification of Cytoskeletal Protein and Axon Injury in PC-12 Cells. Cell Physiol Biochem. 2018;48(3):1088-1098. doi: 10.1159/000491975.
Luo L, Bai R, Zhao Y, Li J, Wei Z, Wang F, Sun B. Protective Effect of Grape Seed Procyanidins against H2 O2 -Induced Oxidative Stress in PC-12 Neuroblastoma Cells: Structure-Activity Relationships. J Food Sci. 2018 Oct;83(10):2622-2628. doi: 10.1111/1750-3841.14349.
Yin D, Zheng X, Zhuang J, Wang L, Liu B, Chang Y. Downregulation of long noncoding RNA Sox2ot protects PC-12 cells from hydrogen peroxide-induced injury in spinal cord injury via regulating the miR-211-myeloid cell leukemia-1 isoform2 axis. J Cell Biochem. 2018 Dec;119(12):9675-9684. doi: 10.1002/jcb.27280.
I round Reviewer’s Point 2: the conclusions about the potential neuroprotective activity of the extracts are overestimated. Indeed, the MTT assay employed to this aim is not enough to demonstrate it. Authors should use more than one test to prove neuroprotection. In other words, the MTT results could be confirmed by another non-colorimetric assay aimed to evaluate the cell viability.
I round Authors’s Response 2: MTT assay is a robust validated and recognised measure of neuronal cell viability via mitochondrial activity. While other assays are available and would be complimentary to the MTT assay, we feel that is sufficient standalone bioassay to measure our desired experimental outcomes, in which we saw statistically significant protective effects of tea extracts on Abeta-mediated cell toxicity.
II round Reviewer’s Response 2: What Authors stated is wrong and extremely insufficient to justify a methodological mistake acknowledged in the whole scientific community. Indeed, it is well-known that employing only one technique to prove whichever effect would be considered insufficient. Moreover, the MTT assay measures indirectly the cell viability via mitochondrial activity, but some chemicals or phytochemicals may change the activity of succinate dehydrogenase or interact with MTT directly, thus interfering with the analysis. Furthermore, the methods employed for the extraction of the plant materials (i.e. solvents) may cause interference in with the assay, resulting in an unreliable viability. Therefore, I absolutely reiterate the need of using more than one test to evaluate cell viability.
II round Authors’ Response 2: We understand and agree with that more technique used will increase the reliability and accuracy of the results. However, we think that MTT assay is sufficiency standalone bioassay to measure our desired experimental outcomes. Firstly, MTT assay is a robust validated and recognised measure of neuronal cell viability via mitochondrial activity. Secondly, the H2O2 treatment results was similar with our previous study on EGCG and all kinds of tea extracts failed to protect neuronal cells against oxidative stress in vitro. Finally, t-bhp treatment also proved that the tea extracts cannot prevent the loss of the PC-12 cell viability. For MTT assay, as demonstrate in 4.4, serum-free media containing 0.25 mg/ml of MTT was added after removing culture media after incubation. Thus, the tea extracts may not able to interact with MTT directly. Furthermore, we observed that all kinds of tea extracts significantly rescue PC-12 cells from Abeta treatment using MTT assay which increased the reliable of MTT assay use in our study. On the basis of above reasons, we do not used another cell viability assay in this study.
III round Reviewer’s Response 2: As I have already stated, it is a methodological mistake employing only one technique to prove whichever effect. Moreover, in the whole scientific community, MTT assay is recognized as a very poor test to prove cytotoxicity, because many agents may interact with MTT directly or interfering with the activity of succinate dehydrogenase, thus profoundly altering the results of the analysis. Therefore, the need to accompany the MTT assay with another test of toxicity.
In addition, EGCG has been shown to protect against mitochondria injury, and greatly increase the activity of succinate dehydrogenase (Hsu et al., 2003; Devika et al., 2008). This may enable the injured cells to reduce MTT and increase the production of formazan by the same number of cells. Finally, some phenolic compounds including kaempferol and EGCG, may reduce directly the reduction of MTT to formazan even in the absence of cells (Bruggisser et al., 2002; Wisman et al., 2008). In this regard, I strongly suggest the Authors to read an important publication about the limitations of MTT-based assays when green tea polyphenols are used (Wang P. et al., 2010).
For all these reasons, I absolutely reiterate the need of using one more test to evaluate cell viability. MTT assay, if it is employed alone, it is extremely insufficient to test cytotoxicity.
Hsu S, Bollag WB, Lewis J, Huang Q, Singh B, et al. (2003) Green tea polyphenols induce differentiation and proliferation in epidermal keratinocytes. J Pharmacol Exp Ther 306: 29–34
Devika PT, Stanely Mainzen Prince P (2008) (-)Epigallocatechin-gallate (EGCG) prevents mitochondrial damage in isoproterenol-induced cardiac toxicity in albino Wistar rats: a transmission electron microscopic and in vitro study. Pharmacol Res 57: 351–357
Bruggisser R, von Daeniken K, Jundt G, Schaffner W, Tullberg-Reinert H (2002) Interference of plant extracts, phytoestrogens and antioxidants with the MTT tetrazolium assay. Planta Med 68: 445–448.
Wisman KN, Perkins AA, Jeffers MD, Hagerman AE (2008) Accurate assessment of the bioactivities of redox-active polyphenolics in cell culture. J Agric Food Chem 56: 7831–7837
Wang P, Henning SM, Heber D. Limitations of MTT and MTS-based assays for measurement of antiproliferative activity of green tea polyphenols. PLoS One. 2010 Apr 16;5(4):e10202. doi: 10.1371/journal.pone.0010202.
I round Reviewer’s Point 3: Authors should search for a possible mechanism of action of the tea extracts.
I round Authors’s Response 3: Studies have been proved that amyloid can cause neuronal cell death by generating ROS. Thus, in our study, we studied the antioxidative effect of tea extract. However, each tea extract failed to rescue PC-12 cells from oxidative stressors. Since many metabolites in tea extracts like catechins, theaflavin, flavonol glycosides and gallic acid, amongst others have been reported to interact with amyloid directly to decrease its toxicity. We opted to focus on the likelihood that these constituent compounds mediated protective effects through this pathway in our Discussion, but also described more generalised cellular protective effects recruited through other component bioactives (e.g. GABA). These aspects have been enhanced in the revised Discussion.
II round Reviewer’s Response 3: I believe that without experimental results explaining any possible mechanism of action, the paper cannot be accepted in this journal.
II round Authors’ Response 3: We firmly agree with that the PC-12 cells will generate ROS and activate some pathways related cell death like caspase under Abeta toxicity. So, in our study, we studied the neuronal cell protective effects of the tea extracts under oxidative stress conditions (t-bhp and H2O2) and Abeta treatment. We found that each tea extract failed to protect neuronal cells from oxidative stress but significantly increased neuronal cell viability following Abeta exposure. Thus, we considered that the key role of tea extract protect PC-12 cells under Abeta exposure in vitro may be the ability to interact with Abeta directly rather than interact with cells. Further ThT and TEM results also supported this point of view. Thus, we think that is not meaningful for us to investigate the protect mechanism of tea extracts related with cells. However, we also agree that the white tea was participated in the cell gene expression regulation to some extent under Abeta toxicity for the high accumulation of GABA or Glutamine. We think further study need to be done to investigate the potential mechanism of action in vitro and in vivo.
III round Reviewer’s Response 3: There are numerous papers demonstrating the capability of tea extracts to reduce the H2O2- or β-amyloid-induced toxicity in neuroblastoma cells through the investigation of their molecular mechanism. In the paper “Neuroprotective and anti-amyloid β effect and main chemical profiles of white tea: comparison against green, oolong and black tea”, the Authors didn’t investigate any mechanism of action to explain the biological activities observed. Therefore, I firmly believe that the study is very much limited and absolutely insufficient to be published in this journal.